# Religious and Non-Religious Workplace Mobbing Victims: When Do People Turn to Religious Organisations?

**DOI:** 10.3390/ijerph191912356

**Published:** 2022-09-28

**Authors:** Jolita Vveinhardt, Mykolas Deikus

**Affiliations:** 1Department of Management, Faculty of Economics and Management, Vytautas Magnus University, 58 K. Donelaičio Street, LT-44248 Kaunas, Lithuania; 2Department of Theology, Faculty of Catholic Theology, Vytautas Magnus University, 58 K. Donelaičio Street, LT-44248 Kaunas, Lithuania

**Keywords:** workplace mobbing, victims, spiritual assistance, religiosity, religious organizations

## Abstract

Researchers’ interest in the impact of religious–spiritual assistance on victims of violence has increased in recent decades; however, factors that are relevant to workplace mobbing victims who seek such assistance remain poorly investigated. The purpose of this study is to highlight the factors that make spiritual assistance of religious organizations acceptable to religious and non-religious workplace mobbing victims. The study involved 463 adults working in Lithuanian organizations, of whom 79.5 per cent indicated that they were religious. ANOVA and Chi-square tests revealed that the significant factors were similarity of personal and religious values, the relation with the person providing assistance, and the circumstances characterising assistance. This study promotes further scientific discussion on the involvement of religious organizations in helping victims of workplace mobbing and explains why religious and non-religious individuals seeking assistance turn to religious organizations. The article presents only a part of the research results of the implemented project.

## 1. Introduction

Although in recent years, research on workplace mobbing has provided a better understanding of this phenomenon and coping strategies [1,2,3], the possibilities of non-profit religious organizations to provide assistance to victims of mobbing at work remain little investigated. However, does this mean that assistance of religious organizations for the victims of mobbing is not relevant? On the one hand, a share of practitioners generally believes that religious needs are in themselves deviations from the norm [4]; therefore, the focus is only on non-religious forms of assistance. On the other hand, a part of researchers who view religious spirituality positively emphasizes a considerable potential of religiosity in dealing with various traumatic experiences [5,6], with particular emphasis on the perspective of spiritual growth in the experienced suffering [7]. It has been noticed that spirituality and religion are important sources of hope and healing for people who have experienced various traumatic experiences [6,8,9]. Therefore, in providing assistance to the person who has experienced a crisis, the focus is on restoring the connection with God and maturing this relationship, which has a dual effect: promotes greater compassion [10] and maintains the ability to find a new way of life after the tragedy in a way that is consistent with the experience of grief and that in the long run, leads to resilience and recovery [11]. Other authors point to the positive effect of victims’ ability to forgive [12,13], the role of religious practices [14], and voluntary assistance of the religious community, which people with various traumas lack [14,15]. In addition, research shows that community religious practices, works for the loved one, volunteering and other activities significantly reduce the person’s experienced pain, promote integration and growth [16], and that religious individuals tend to seek not only psychotherapy but also assistance in their religious communities [17,18,19]. Although advisors working in religious communities are not formal mental help professionals [17], separate elements of psychotherapy are applied in practice [20], and the provided spiritual assistance is based on a holistic model that takes into account biological, psychological, social and religious–spiritual aspects of human existence [21,22]. In other words, according to this model, the vertical dimension that is understood as God-oriented religious life and the horizontal dimension encompassing the person‘s physiological, psychological and social relations aspects are equally important. It is significant in this context that the search for religious–spiritual assistance after experiencing various traumas is related to personal growth [21]. In other words, although religious–spiritual assistance is not treated as some alternative to psychotherapy, its efficiency is confirmed in empirical research, while the need for spiritual assistance is acknowledged by both religious and non-religious individuals [23]. However, the studies reviewed above do not directly address religious–spiritual assistance to individuals who have experienced various psychological traumas in the workplace. It is also not clear what aspects characterising voluntary religious organizations are or may be relevant and attractive to employees who have experienced workplace mobbing or other types of attacks. Therefore, this study aims to highlight the factors due to which spiritual assistance of religious organizations is acceptable to religious and non-religious workplace mobbing victims. In order to achieve the set aim, two problem questions are formulated:

Q1: What motives and circumstances encourage workplace mobbing victims to seek spiritual assistance in religious organizations?

Q2: How do the motives of religious and non-religious persons who have experienced workplace mobbing differ in seeking religious–spiritual assistance?

In order to answer the above questions, first, we review the scientific literature, make assumptions, and test them empirically. We perform quantitative research on employee experiences related to workplace mobbing. This shows the relative scope of possible voluntary activities. Second, we analyse the circumstances of the search for religious–spiritual assistance, which can reveal both the current situation and show further directions for the development of the service. Finally, we compare the motives of persons who have experienced and who have not experienced workplace mobbing, which inspires the search for religious–spiritual assistance.

Considering the analysed problem, the scientific literature search included publications in both the humanities and social sciences (theology, psychology, management, and other fields) in journals referenced in the Web of Science Core Collection and/or Scopus databases. The search was performed by entering keywords “workplace mobbing”, “spiritual assistance”, “pastoral care”, “voluntary”. Given the lack of scientific literature on the analysed topic, we sought that the findings of this study filled the gap of this type of scientific research, i.e., we aimed to confirm the importance of religious values and to explain why persons who had experienced workplace mobbing needed such assistance. It should be emphasized that this article presents only a part of the results of the conducted study.

## 2. Literature Review

*Religious–spiritual assistance* includes the horizontal level (helping the person to get closer to his or her suffering in earthly life) and the vertical level, which is made up of the person’s religious and/or spiritual approaches and beliefs [24]. In other words, a holistic approach to the individual is used, where the person’s well-being is understood inseparably from the social and transcendent dimensions [24,25,26]. This type of assistance is provided by employing the resources of religious faith and using interventions grounded on advances in the science of psychology [27,28,29]. Summarising typologies of religious–spiritual assistance practices, Marques (2021) [30] distinguishes the following key practices used in the Catholic context: “accompanying” and “leading to other religious mediators” (p. 6). Other less frequently used practices include meditation on the lives of the saints, liturgical celebrations, pilgrimages, etc.

*Workplace mobbing* is seen as a form of behaviour causing extreme stress, which includes persistent insulting and harassment, is directed at one person, and lasts for a specific period of time [31,32]. In order to distinguish mobbing from other forms of unethical behaviour in the workplace, Leymann (1996) [33] has proposed that mobbing should be considered an attack that lasts for at least six months, with the attacks recurring at least once a week. This criterion enables to distinguish mobbing from other types of destructive relationships in the workplace, which do not necessarily distinguish themselves by such intensity and duration or when one person harasses. Another characteristic feature of mobbing is that the victim usually feels lonely, acquires a disadvantaged position, and is unable to defend oneself [31,34]. Therefore, workplace mobbing is described as a traumatic process that severely damages the victim’s mental and physical health [32]. In this study, the authors follow the position that destructive behaviour is the one that falls outside the above-mentioned concept of workplace mobbing.

*Religiosity in choosing assistance*. The person’s religiosity that is based on value congruence is one of the main motives for seeking help from religious organizations. Although religion occupies an important place in the culture of societies, the level of religiosity varies from state to state [35,36]; therefore, the issue of using religious sources of assistance while seeking to overcome emerging crises remains relevant. In this context, the study among Catholics in the United States, conducted by Harris et al. (2008) [21], demonstrated that the search for spiritual assistance was associated with greater religiosity, while the relationship with God and other people was seen as a source of comfort and support in order to cope with experienced traumas. These trends are also confirmed by other studies showing that religious individuals preferred religious assistance when looking for ways to alleviate the psychological suffering they were experiencing [17,19]. In addition, it has been established that there is a relationship between religious behaviour, strength of faith, spirituality, and religious coping [37] and that the search for religious assistance does not depend on the age of the person seeking it [17,19]. Crosby and Varela (2014) [38] believe that people for whom religiosity serves as a means of overcoming existential anxiety perceive a special connection with their deity and believe that this relationship provides unique protection in life. Therefore, the person with such beliefs understands emotional experience as a spiritual rather than a mental health problem. In such cases, reluctance to seek mental health professional’s assistance can be strong. In addition, Brenner et al. (2018) [39] pointed out that the choice of the source of help may be influenced by factors such as religious commitment and stigmas. On the other hand, the study conducted by Fox et al. (2020) [40] showed that the choice of religious assistance was influenced by religious and spiritual variables as well as the spiritual bypass where the person avoids painful psychological experiences and uses spirituality as defence. Although no studies directly addressing mobbing/bullying victims’ search for religious–spiritual assistance could be found, the research reviewed above suggests that the values of a religious victim experiencing co-workers’ attacks and the victim’s relation with the religious community can play an important role in seeking religious–spiritual assistance.

*Theories explaining help-seeking*. Certain answers to questions as to why people who have found themselves in crises seek religious–spiritual assistance can be obtained in several theories. One of these is the theory of planned behaviour, which, according to Ajzen (2020) [41], was successfully used to explain and predict behaviour in many behavioural domains ranging from the physical activity to drug use, from recycling to choice of travel, etc. This theory takes into account attitudes toward the behaviour, subjective norms, perceived behavioural control and intentions [41]. According to Taylor and Kuo (2019) [42] the key component of the theory of planned behaviour is intention, which is defined as the extent to which the person is willing to make an effort in order to perform the behaviour. In addition, important factors for help-seeking are social pressure and beliefs. Findings of the research conducted by Lefevor et al. (2021) [43] showed that the behaviour of clergy (the extent to which they talk about help) had determined the sense of control over the ability to seek help and a more positive attitude towards the search for assistance.

Social comparison theory is another approach that can be used to explain the mechanism of seeking help or support in different contexts (e.g., [44,45,46]. It can help to define reference groups against which people compare their experiences when making decisions about help-seeking [46]. According to Gerber (2018) [47], social comparison consists of several parts: “who people compare with, why they compare, the effects of those comparisons, and who is likely to compare”. Studies also take into account variables such as gender, age, self-esteem, values, etc. Normally, people tend to compare when something is not known and stop comparing when more solid information is available.

*Information on the provided assistance*. Although help-seeking depends on the victim’s personal traits, attention is also paid to contextual factors such as information about assistance opportunities and readiness of potential sources of assistance to provide assistance [48,49]. In this context, several important factors need to be taken into account, such as trust and the ways of disseminating information. According to Sheikhi et al. (2021) [50], leaders of religious organisations are those individuals who feel responsible for the community’s positive mental health state and who are considered a trustworthy source of information. Faithful members of the community find it easier to accept the information received from them and they follow what is said. Moreover, declining church attendance and the proliferation of social networks have not significantly undermined the role of official sources of religious information and knowledge [51]. At the same time, the ways in which consumers of religious products are reached and influenced in one way or another are also changing [52,53]. That is, from the marketing perspective, religious organisations have excellent opportunities to present the religious–spiritual assistance service. Although the specific formula of the exchange is treated as commercial, these exchanges can be measured in spiritual categories, since the Church meets the religious needs of its members [52]. However, although there is a lack of the most recent research investigating the impact of disseminating information about religious–spiritual assistance on the choice of such assistance, some evidence demonstrates that in practice, this subject sometimes receives insufficient attention [54] or that assistance is generally only made available to the members of a particular denomination [55].

## 3. Materials and Methods

*Participants*. Data were collected by surveying persons working in Lithuania. No official statistics on the number of religious persons of working age have been found, but according to the data of the last census that took place in 2011, 2.5 million residents considered themselves religious (Roman Catholics, Orthodox believers, Evangelical Lutherans, Evangelical Reformed believers, Sunni Muslims, etc.) (Population and housing censuses in Lithuania). According to the data of the Lithuanian Department of Statistics, in 2019, the average number of employees in the national economy, including sole proprietorships, was 1 million 287.9 thousand (Labour market in Lithuania). This survey involved 463 adult working persons. According to the Lithuanian working population, at 95 per cent confidence and 5 per cent error, a representative sample must consist of at least 384 respondents.

*Procedure*. Due to the quarantine announced in the country, physical contacts were restricted; therefore, an online survey was conducted: the questionnaire was placed on an online survey platform but was not publicly announced and available. Using data from the Register of Legal Entities, the list of 1500 email addresses was compiled from randomly selected publicly available contacts of employees working public and private sector enterprises. Invitations to participate in the survey were sent to these addresses. To complete the questionnaire, features prohibiting to answer questions from the same electronic device more than once were set. Answers with the same response ratings were also blocked. The feature ‘mandatory answer’ was enabled to all questions given in the questionnaire; therefore, there were no incompletely filled-in questionnaires.

Research participants were introduced to the aims of the survey, their anonymity and confidentiality were guaranteed, and it was explained that they could terminate their participation in the survey at any time. Pursuant to Article 1.64 of the Civil Code of the Republic of Lithuania, the informed consent is given in two ways: active (written, verbal, by actions) and passive (non-objection, non-requirement to terminate, implicit actions). Therefore, the completion of the questionnaire, i.e., voluntary provision of answers, was considered an expression of such consent. Research participants were informed that the data collected during the survey would be used exclusively for scientific purposes.

*Measures*. The study was conducted using the questionnaire “Motives of Persons Aggrieved at Work for Seeking Spiritual Assistance (MP-SSA-40)” [56]. This questionnaire was compiled after analysing previously conducted studies that investigated individual aspects close to our study. It was found that the instrument used was valid and reliable [57]. The questionnaire consists of 4 scales and 7 subscales covering 40 items. The reliability of the subscales was established by using the Cronbach’s alpha coefficient: Destructive actions, α = 0.86; Causes of destructive actions, α = 0.69; Personal values, α = 0.89; Values of religious organisations, α = 0.89; Knowledge of assistance provided by religious organisations, α = 0.90; Motives determined by the relation, α = 0.93; Motives determined by circumstances, α = 0.90. Intercorrelation relationships at the scale level show a strong relationship between service awareness and values (0.629, *p* = 0.000) and between motives and values (0.740, *p* = 0.000). A moderately strong relationship was found between motives and service awareness (0.586, *p* = 0.000). Very weak relationships were recorded between destructive relationships, service awareness (0.094, *p* = 0.044) and motives (0.186, *p* = 0.000), but the established relationship is statistically reliable. However, no relationship between values and destructive relationships (*p* = 0.052) was found.

With regard to experiences of co-workers’ destructive behaviour, the participants of this study were divided into three groups: (1) those who experienced mobbing (i.e., experienced co-workers’ destructive actions when (1.1) attacks recurred at least once a week, (1.2) continued for more than half a year, (1.3) and the actions were directed by more than one person; i.e., by a group of persons); (2) those who experienced destructive behaviour but did not experience mobbing (i.e., experienced co-workers’ destructive actions when (2.1) attacks recurred less than once a week, (2.2) lasted for several months, (2.3) destructive actions were taken by one person, usually without accomplices); (3) those who did not experience either mobbing or destructive behaviour.

## 4. Results

### 4.1. Demographic Characteristics

The results show that the number of religious and non-religious persons in percentages in principal corresponds to the trends identified in the census (differs by about 2 per cent). A total of 79.5 per cent of respondents indicated that they believed in God, of which the majority were women—86.4 per cent (men—69.8 per cent) (Table 1). The largest share of respondents that have fallen into the sample (72.8 per cent) indicated that they belonged to the Roman Catholic community, which almost corresponds to the trends seen in the Lithuanian census (i.e., in the census of 2011, 77.2 per cent of the population attributed themselves to Roman Catholics). Less than a fifth of respondents did not belong to any faith community or indicated that they were agnostics. The research results show that a share of those who attributed themselves to non-religious persons or agnostics still has contacts with the houses of worship of religious communities, which becomes evident from the responses about visiting them. Despite the significant number of religious persons, the liturgical rites are attended by less than a tenth of respondents, and barely 5.6 per cent actively participate in the activities of religious communities.

### 4.2. Experiences of Workplace Mobbing and Destructive Relationships at Work

For the purposes of this study, another criterion was experiences of workplace mobbing and destructive relationships at work, which are not attributed to it. Almost half (49.7 per cent) of respondents indicated that they had not experienced destructive behaviour of employees, while the behaviour attributable to workplace mobbing was noted by 16.8 per cent of survey participants. The remaining part—33.5 per cent—experienced destructive relationships, i.e., when destructive type attacks recurred less than once a week or for less than half a year. Meanwhile, the percentage of men and women who have not experienced destructive behaviour in principal does not differ, but there were more persons who experienced destructive relationships in the women’s group (Table 1).

The Student’s *t*-test and ANOVA test results (Table 2, Table 3 and Table 4) showed several significant trends that help to understand the role of motives and circumstances in seeking assistance from religious organizations (Q1). First, approval of values, awareness of the spiritual assistance service, and motives for help-seeking were more pronounced among religious persons, and the *t*-test verification results show, as it might have been expected, statistically significant differences (Table 2).

Second, manifestation of destructive relationships, values and motives statistically significantly differs depending on the experience of co-workers’ negative behaviour. Considering the results, the said features are most reflected in the group of persons who have experienced mobbing; and least, in the group of persons who have not experienced such behaviour. Awareness of the service is similar in all compared groups, which indicates that experience does not have a significant effect (Table 3).

Third, at the sub-scale level, manifestation of destructive relationships at work and motives differs statistically significantly depending on the experience of co-workers’ destructive behaviour. The differences are shown by the results of the analysis of variance, and in those cases, the contrast statistically significantly differs from zero. Hence, in the aspects of values (personal and of religious organizations) and awareness of services, no statistically significant differences between the means were identified and in the said cases the manifestation of features is similar (Table 4).

Further, answers were sought to the question, how the motives of religious and non-religious persons who had experienced workplace mobbing differed in seeking religious–spiritual assistance (Q2). On the other hand, First of all, at the sub-scale level, it was checked how respondents’ values, motives, knowledge of spiritual assistance, destructive actions and their causes unfolded depending on the nature of negative behaviour experiences and the belief/disbelief in God. Research participants’ responses given below are divided into three groups: responses of those who experienced destructive behaviour, of those who experienced mobbing, and of those who did not experience any destructive actions from co-workers. The results of the Chi-square test show that there is no statistically significant dependence only with regard to the values of religious organizations. The comparison of respondents’ answers highlighted that the persons who experienced mobbing put more emphasis on both actions and their causes than those who experienced only destructive behaviour. Personal values were more important to them as well as the motives determined by circumstances and the relation with the person providing assistance (Table 5).

The Chi-square test results show that there is no statistically significant dependency only in the scales of destructive actions and causes. As expected, the role of values was most evident in the group of religious persons who experienced destructive relationships. The difference between personal values and values of religious organizations is insignificant, amounting to only three-tenths of a per cent. It should be noted that in the group of religious persons, more significance was attached to knowledge of the provided assistance, circumstances, and interpersonal relationships. When comparing what is more important—the relation with the person providing assistance or circumstances—the latter overweight. Besides, certain trends that have emerged in the group of non-religious persons are also noteworthy. First, quite a high level of approval of values that are traditionally associated with religious values and are declared by religious organizations was established. Second, slightly more than half of respondents emphasized the role of circumstances, although the number of persons to whom the relation with the person providing assistance was significant was five times less (Table 6).

## 5. Discussion

The results of this study explain what motives and circumstances can encourage victims of workplace mobbing to seek religious–spiritual assistance (Q1). On the one hand, the results of the study show that religious individuals are more motivated to seek religious–spiritual assistance and that values (both personal and values of religious organizations) and awareness of the service are significant factors. For example, Dunaetz et al. (2020) [58] found that emotional commitment to one’s church positively correlated with value congruence. The importance of congruence of religious values has also been noted in the relationship of clients and Jewish Ultra-Orthodox therapists with help-seeking believers [59]. Other studies on the relationships between non-religious organizations and their service users show an important role played by value congruence [60]. Based on the planned behaviour theory, values are attributed to key factors (along with personality traits, intelligence, demographic characteristics) and “they are assumed to influence intentions and behavior indirectly by affecting behavioral, normative, and/or control beliefs” [41] (p. 5). On the one hand, people who have experienced workplace mobbing form that group of individuals for which religious values are most relevant and which may be most motivated to seek religious spiritual assistance. From the perspective of the social comparison theory, victims seeking assistance could have compared their personal experiences with those of other persons (through direct contacts and stories in the media). However, in this case, awareness of the spiritual assistance service was not that circumstance which would distinguish any group in terms of workplace mobbing experiences from others (those who have experienced only destructive relationships or those who have not suffered in the workplace). This shows that victims were not concerned about information about this type of help more than others. It is natural that religious persons have closer contacts with their religious community and receive more information about various possibilities of assistance, but it is significant that non-religious persons had such information as well. Considering coinciding personal values and values of religious organizations and quite a large share of respondents who take into account occurring circumstances and the personal relation with assistance providers, this can be a weighty basis for expanding the scope of assistance. All the more so as assistance is understood not only as direct counselling but also as help, support of the community for victims outside the religious community [61,62].

On the other hand, the motives of religious and non-religious individuals who experienced workplace mobbing for seeking religious–spiritual assistance differed (Q2). First, in the case of religious persons, the major emphasis is placed on contextual factors such as values declared by religious organizations and circumstances (e.g., competence of the person providing assistance, recommendations, non-receipt of assistance from other professionals, reward for the service). Second, the personal relation with the person providing assistance remains particularly important too. In other words, the mere fact that the victim believes in God, belongs to a particular religious community does not guarantee that assistance will be sought. It is significant that these circumstances are also relevant to some non-religious individuals. Our study demonstrates that not only the value-based connection with a religious organization, but also the severity of experience is important for seeking religious–spiritual assistance. That is, the more desperate the situation the person experiences, the higher the probability that the person will turn to a religious organization. Studies show that the need for religious–spiritual assistance is highly pronounced in the face of serious illness or death when in addition to physical, psychological and social pain, persons also experience severe spiritual pain [22,63]. Similarly, the feeling of hopelessness and stress experienced during mobbing can be equated to a confrontation with death [32], all the more so that the insecurity associated with job loss often becomes a reason for suicide [64]. In other words, in addition to explanations of the influence of some cultural and social variables on religious–spiritual help-seeking [19], the severity of mobbing experience can be understood as a stimulus to seek assistance that would meet the specific needs of the religious victim. This broadens the understanding of the motives for seeking religious–spiritual assistance and at the same time shows that in addition to assurance of psychological assistance, victims of mobbing may also benefit from support consistent with the individual’s religious beliefs. That is, Thus, religious organizations have some potential to provide assistance to non-religious persons seeking it as well.

*Limitations and Prospects*. The study has several limitations. The strength of religiosity, which might have influenced respondents’ choices, was not assessed separately. This study also did not investigate social and demographic factors that might have impact on seeking religious–spiritual assistance. Previous research has shown that gender and belonging to certain age groups are important factors explaining workplace mobbing victims’ motives for seeking assistance in general [65,66]; therefore, other studies should also assess how these factors influence turning to religious organizations. This study is quantitative and shows general trends, but some motives of employees seeking assistance may have remained unevaluated. These could be detailed by in-depth interviews with persons experiencing negative behaviour of co-workers. The results of the study show that non-religious victims of mobbing may also consider the opportunity to seek assistance from religious organizations. The qualitative research would help to understand how such intentions may be related to, for example, contextual actions such as living among religious persons, availability and quality of psychological assistance, general trust in therapy, societal attitudes related to psychological assistance, etc. Interventions applied in pastoral practice should also be evaluated. This study only partially considered the insights of the theories of planned behaviour and social comparison; therefore, in the future, their application should be more comprehensively evaluated while analysing the behaviour of mobbing victims seeking religious–spiritual assistance. All the more so that, to our knowledge, no such studies have been conducted to date. It was also not analysed how the search for help differed between representatives of different denominations. It is significant that the results of this study show that information about the opportunities for spiritual assistance also reaches non-religious persons who, depending on the occurring situation, may seek assistance from the religious organization [62]. Therefore, in further research, it would make sense to conduct a more thorough investigation on the circumstances in which persons belonging to this group would seek such assistance. This study well represents the situation in Lithuania, but the religious context in other countries may differ; therefore, more research should be conducted to achieve broader generalizations.

## 6. Conclusions

As the possibilities of religious organizations to provide assistance to the victims of workplace mobbing have so far been little investigated, this study provides new knowledge that can be useful for both the development of this type of assistance and new research. It fills the gap in such literature, confirms the importance of religious values (compared with other factors), and explains why people who have experienced workplace mobbing tend to seek religious–spiritual assistance. On the one hand, we seek to promote further scientific discussion on the involvement of religious organizations in the provision of assistance to the victims of both workplace mobbing and destructive relationships at work. On the other hand, we highlighted factors whose evaluation in practice would enable religious organizations to organize volunteering better, this way providing more opportunities for victims at work to receive additional spiritual assistance. We also demonstrate why religious organizations should not limit themselves to the members of their community when organizing voluntary activities.

## Figures and Tables

**Table 1 ijerph-19-12356-t001:** Religiosity and experiences of co-workers’ destructive behaviour with regard to gender.

	Gender	Males *N* = 183		Females *N* = 280		Total *N* = 463	
Religiosity and Relationships		*N*	*%*	*N*	*%*	*N*	*%*
Religiosity	Religious persons	126	68.9	242	86.4	368	79.5
	Non-religious persons	57	31.1	38	13.6	95	20.5
Experiences of co-workers’ destructive behaviour	Did not experience	99	54.1	131	46.8	230	49.7
	Experienced mobbing *	25	13.7	53	18.9	78	16.8
	Experienced destructive behaviour **	59	32.2	96	34.3	155	33.5

* Experienced mobbing when co-workers’ attacks recurred at least once a week, lasted for more than half a year, a group of individuals attacked; ** Experienced destructive behaviour when co-workers’ attacks recurred less than once a week, lasted for several months, not necessarily a group of individuals attacked.

**Table 2 ijerph-19-12356-t002:** The *t*-test verification by believing in God.

Scales	Believe in God *N* = 368		Do Not Believe in God *N* = 95		*T*-Test Verification Results	
	*Mean*	*SD*	*Mean*	*SD*	*t*	*p*
Destructive relationships	1.82	0.69	1.66	0.59	2.016	0.044 *
Values	3.33	0.76	2.02	0.61	15.427	0.0001 **
Awareness of the service	2.88	0.84	1.93	0.83	9.742	0.0001 **
Motives	3.23	0.83	2.09	0.83	11.964	0.0001 **

* statistical significance level α = 0.05; ** statistical significance level α = 0.01.

**Table 3 ijerph-19-12356-t003:** The ANOVA test at the scale level by the nature of experience.

Scales	Did Not Experience Destructive Behaviour *N* = 230		Experienced Mobbing *N* = 78		Experienced Destructive Behaviour *N* = 155		ANOVA Test Results	
	*Mean*	*SD*	*Mean*	*SD*	*Mean*	*SD*	*F*	*p*
Destructive relationships	1.38	0.36	2.64	0.62	1.95	0.60	196.293	0.0001 **
Values	2.96	0.89	3.23	0.86	3.12	0.94	3.286	0.038 *
Awareness of the service	2.61	0.91	2.88	0.94	2.69	0.92	2.497	0.083
Motives	2.82	0.98	3.38	0.81	3.08	0.91	11.498	0.0001 **

* statistical significance level α = 0.05; ** statistical significance level α = 0.01.

**Table 4 ijerph-19-12356-t004:** The ANOVA test at the subscale level by the nature of the experience.

Subscales	Did Not Experience Destructive Behaviour *N* = 230		Experienced Mobbing *N* = 78		Experienced Destructive Behaviour *N* = 155		ANOVA Test Results		Contrast (−1, 2, −1)	
	*Mean*	*SD*	*Mean*	*SD*	*Mean*	*SD*	*F*	*p*	*Cr*	*Cr. p*
Destructive actions	1.51	0.47	2.84	0.69	2.19	0.75	150.131	0.0001 **	1.977	0.0001 **
Causes of destructive actions	1.26	0.40	2.43	0.83	1.71	0.71	114.041	0.0001 **	1.901	0.0001 **
Personal values	2.88	0.98	3.16	0.94	3.11	0.98	3.816	0.055	0.335	0.167
Values of religious organizations	3.04	0.90	3.31	0.90	3.14	0.98	2.477	0.085	0.433	0.062
Knowledge of assistance provided by religious organizations	2.61	0.91	2.88	0.94	2.69	0.92	2.497	0.083	0.463	0.068
Motives determined by circumstances	3.07	1.00	3.66	0.85	3.38	0.92	12.887	0.0001 **	0.881	0.0001 **
Motives determined by the relation	2.57	1.09	3.09	1.00	2.77	1.05	7.443	0.001 **	0.850	0.001 **

*Cr*—Contrast value; *Cr. p*—Contrast value, *p*. ** statistical significance level α = 0.01.

**Table 5 ijerph-19-12356-t005:** The Chi-square test by different experiences.

Subscales	No/Yes ^1^	Did Not Experience Destructive Behaviour *N* = 230		Experienced Mobbing *N* = 78		Experienced Destructive Behaviour *N* = 155		Chi-Square Test Results	
		*N*	*%*	*N*	*%*	*N*	*%*	*χ* ^2^	*p*
Destructive actions	No	209	90.9	17	21.8	90	58.1	139.409	0.0001 **
	Yes	21	9.1	61	78.2	65	41.9		
Causes of destructive actions	No	227	98.7	41	52.6	131	84.5	104.598	0.0001 **
	Yes	3	1.3	37	47.4	24	15.5		
Personal values	No	48	20.9	16	20.5	17	11.0	6.882	0.032 *
	Yes	182	79.1	62	79.5	138	89.0		
Values of religious organizations	No	50	21.7	14	17.9	21	13.5	4.155	0.125
	Yes	180	78.3	64	82.1	134	86.5		
Knowledge of assistance provided by religious organizations	No	136	59.1	34	43.6	60	38.7	16.836	0.001 **
	Yes	94	40.9	44	56.4	95	61.3		
Motives determined by circumstances	No	62	27.0	8	10.3	26	16.8	12.107	0.002 **
	Yes	168	73.0	70	89.7	129	83.2		
Motives determined by the relation	No	143	62.2	29	37.2	79	51.0	15.647	0.001 **
	Yes	87	27.8	49	62.8	76	49.0		

^1^ No—non-religious persons; Yes—religious persons. * statistical significance level α = 0.05; ** statistical significance level α = 0.01.

**Table 6 ijerph-19-12356-t006:** The Chi-square test by belief in God.

Subscales	No/Yes ^1^	Believe in God *N* = 368		Do Not Believe in God *N* = 95		Chi-Square Test Results	
		*N*	*%*	*N*	*%*	*χ* ^2^	*p*
Destructive actions	No	247	67.1	69	72.6	1.069	0.304
	Yes	121	32.9	26	27.4		
Causes of destructive actions	No	314	85.3	85	89.5	1.090	0.296
	Yes	54	14.7	10	10.5		
Personal values	No	43	11.7	38	40.0	41.941	0.0001 **
	Yes	325	88.3	57	60.0		
Values of religious organizations	No	42	11.4	43	45.3	57.725	0.0001 **
	Yes	326	88.6	52	54.7		
Knowledge of assistance provided by religious organizations	No	151	41.0	79	83.2	53.599	0.0001 **
	Yes	217	59.0	16	16.8		
Motives determined by circumstances	No	49	13.3	47	49.5	60.067	0.0001 **
	Yes	319	86.7	48	50.5		
Motives determined by the relation	No	166	45.1	85	89.5	59.872	0.0001 **
	Yes	202	54.9	10	10.5		

^1^ No—did not experience destructive relationships; Yes—experienced destructive relationships. ** statistical significance level α = 0.01.

## Data Availability

Data available on request due to restrictions.

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
