# Peer review of "Religious and Non-Religious Workplace Mobbing Victims: When Do People Turn to Religious Organisations?"

_ijerph, 2022, doi:10.3390/ijerph191912356_

Round 1
Reviewer 1 Report
The present paper sought to describe the motivations among religious and non-religious individuals’ decisions to turn toward religious organizations for assistance in light of experienced workplace mobbing. Below I detail some points for concern or additional clarification upon my reading of the paper.
1. Can a definition of religious-spiritual assistance, even if brief, be provided earlier in the piece? The differentiation between horizontal and vertical is provided at the top of the literature review, but clarifying terms within the introduction can aid readers less familiar with that practice.
2. Why did the sampling include employees who did not experience any type of workplace mistreatment? I acknowledge the distinction between those who experienced mistreatment at levels of mobbing and those who experienced less severe levels, but given the described research aims, it seems that only focusing on those with such experiences would have made sense to be reflected within the sampling approach.
3. Were any qualitative measures used to attempt to understand motives more deeply? I understand you used a particular scale to capture participant motives, but it seems like a missed opportunity to hear participants’ own descriptions of why they engaged in a particular behavior, unbounded by the constraints of a quantitative scale.
4. Within your discussion of results, it would be helpful to tie your findings back to your original research questions- I found myself having to check back and infer for myself how the statistics I’m reading potentially inform the questions guiding the study.
5. Study implications could have been made clearer. For instance, the finding that religious individuals are more likely to seek religious spiritual assistance doesn’t strike me as particularly novel or contributive- greater clarity surrounding contributions can help underscore the value of this work.
I hope these comments prove useful, and best of luck with your continued research!
Author Response
Review 1 (R1)
R1: The present paper sought to describe the motivations among religious and non-religious individuals’ decisions to turn toward religious organizations for assistance in light of experienced workplace mobbing. Below I detail some points for concern or additional clarification upon my reading of the paper.
- Can a definition of religious-spiritual assistance, even if brief, be provided earlier in the piece? The differentiation between horizontal and vertical is provided at the top of the literature review, but clarifying terms within the introduction can aid readers less familiar with that practice.
Authors (1): We sincerely thank you for the shortcomings indicated in your review, to which we have responded.
Considering your comments, we have added brief explanations in the introduction (all additions are marked with a different colour in the manuscript).
- Why did the sampling include employees who did not experience any type of workplace mistreatment? I acknowledge the distinction between those who experienced mistreatment at levels of mobbing and those who experienced less severe levels, but given the described research aims, it seems that only focusing on those with such experiences would have made sense to be reflected within the sampling approach.
Authors (2): By including persons who had not experienced negative behaviour in the sample, we pursued several goals. First, to show the proportions of employees experiencing and not experiencing negative relationships. Second, in our opinion, this provides an opportunity to compare how, for example, certain motives of persons who are in different situations manifest themselves.
- Were any qualitative measures used to attempt to understand motives more deeply? I understand you used a particular scale to capture participant motives, but it seems like a missed opportunity to hear participants’ own descriptions of why they engaged in a particular behavior, unbounded by the constraints of a quantitative scale.
Authors (3): In this case, only a quantitative research approach was used, as we sought to statistically test certain assumptions. Also, considering your comment, we are supplementing the section on the limitations of research.
- Within your discussion of results, it would be helpful to tie your findings back to your original research questions- I found myself having to check back and infer for myself how the statistics I’m reading potentially inform the questions guiding the study.
Authors (4): We have supplemented the parts Research Results and Discussion, which we hope should make reading easier (marked with a different colour).
- Study implications could have been made clearer. For instance, the finding that religious individuals are more likely to seek religious spiritual assistance doesn’t strike me as particularly novel or contributive- greater clarity surrounding contributions can help underscore the value of this work.
Authors (5): We took the comment into consideration and supplemented the manuscript.
I hope these comments prove useful, and best of luck with your continued research!
Authors (6): Thank you for your valuable observations that improved the quality of our manuscript.

Reviewer 2 Report
The apparent aim of this paper is to assess the extent, and in what manner, religious agencies and counsellors can intervene on behalf of or with workplace bullied individuals (and with their workplace peers, and management) in order to diminish such bullying, and to counsel the bullied individuals.
This is a difficult paper to review, and raises many questions which are not answered satisfactorily.
First “workplace mobbing” does occur, but is better described as “group bullying in the workplace or educational institution”:
https://smallbusiness.chron.com/mobbing-workplace-43426.html
I would advise the use of a different term than “mobbing”, which is an ethological concept involving group physical attacks on an unwanted threat, by a weaker group on a perceived interloper or competitor.
Using the term “mobbing” for workplace bullying requires much more explanation and definition of the concept, ideally with some case histories derived from the research project. Very little indication is given of why individuals in this study were subjected to group bullying, nor to what degree. If counsellors (religious or secular) are to intervene on behalf of the bullied individuals it is important to know how they did this, or would do this. Did they intervene with management on behalf of the bullied individuals? What were the outcomes? What are the mental health profiles of the bullied individuals? Did intervention by religious counsellors improve this mental health? I tried hard to understand the convoluted prose presentation of this article, but could not get any sense of this. Did the research actually collect data in these crucial areas?
Were individuals subjected to group bullying because of their manifest religious identity (e.g. hijab-wearing Muslims; cross-wearing Christians)?
Will religious intervention simply counsel the individual to “bear the cross” imposed upon them; or will the religious agency intervene in order to end the bullying, whatever the reason?
Qualitative case studies would illustrate the various stresses placed on individuals, and strategies engaged in by the religious counsellors. Without this information, I don’t find this article helpful in understanding the nature of group bullying within the culture of Lithuania, the nature of interventions possible, and their effectiveness.
In what kind of agency (e.g. public or private) does group bullying occur, and to what extent, and why? If the authors have any information in this regard, they should present it. How many respondents belonged to religious minorities (e.g. as Jews, Muslims, Jehovah’s Witnesses etc) – were they particularly likely to experience group bullying?
A questionnaire is mentioned: the English-language translation should be appended. What are the “values” held by respondents? This could be a crucial concept, which is not elaborated. How many people who are “religious” but who rarely attend church, hold positive values regarding self and others? Can they be co-opted as resource people to counter workplace bullying? How do employers/directors regard intervention in the workplace by religious agencies?
My conclusion is that merely as a review of literature, and as a descriptive survey of religious beliefs and experience of workplace bullying, this article does not offer much advance on previous studies. We need to know more exactly how religious agencies and counsellors can intervene on behalf of bullied individuals, and what the outcomes have been, or are likely to be.
Author Response
Review 2
R2: The apparent aim of this paper is to assess the extent, and in what manner, religious agencies and counsellors can intervene on behalf of or with workplace bullied individuals (and with their workplace peers, and management) in order to diminish such bullying, and to counsel the bullied individuals.
This is a difficult paper to review, and raises many questions which are not answered satisfactorily.
First “workplace mobbing” does occur, but is better described as “group bullying in the workplace or educational institution”:
https://smallbusiness.chron.com/mobbing-workplace-43426.html
I would advise the use of a different term than “mobbing”, which is an ethological concept involving group physical attacks on an unwanted threat, by a weaker group on a perceived interloper or competitor.
Using the term “mobbing” for workplace bullying requires much more explanation and definition of the concept, ideally with some case histories derived from the research project. Very little indication is given of why individuals in this study were subjected to group bullying, nor to what degree. If counsellors (religious or secular) are to intervene on behalf of the bullied individuals it is important to know how they did this, or would do this. Did they intervene with management on behalf of the bullied individuals? What were the outcomes? What are the mental health profiles of the bullied individuals? Did intervention by religious counsellors improve this mental health? I tried hard to understand the convoluted prose presentation of this article, but could not get any sense of this. Did the research actually collect data in these crucial areas?
Authors’ answer (1). It occurs in the literature that the term workplace bullying (talking of group bullying) is used as a synonym for workplace mobbing, and this leads to debates. Both in the theoretical part and while drawing up the questionnaire, we referred to the tradition formed by H. Leymann, according to which a distinction is made between “workplace bullying” and “mobbing”. We agree that the reader may be confused; therefore, we have made some additions to the manuscript (all additions are marked with a different colour). We hope that they will allow the reader to better understand the phenomenon being studied.
R2: Were individuals subjected to group bullying because of their manifest religious identity (e.g. hijab-wearing Muslims; cross-wearing Christians)?
Authors’ answer (2). Although faith was indicated as one of the reasons for attacking in the questionnaire, our study did not focus on persecution on the basis of a specific religious identity, as this could be the object of another study.
R2: Will religious intervention simply counsel the individual to “bear the cross” imposed upon them; or will the religious agency intervene in order to end the bullying, whatever the reason?
Authors’ answer (3). We agree that this is an important issue, but in this case we investigated not the pastoral response, but rather the motives that, taking into account different experiences, encourage to seek help from religious institutions. We see the gap of such knowledge, which we tried to fill in and accordingly supplemented the discussion section of the manuscript:
“Our study demonstrates that not only the value-based connection with a religious organization, but also the severity of experience is important for seeking religious-spiritual assistance. That is, the more desperate the situation the person experiences, the higher the probability that the person will turn to a religious organization. Studies show that the need for religious-spiritual assistance is highly pronounced in the face of serious illness or death when in addition to physical, psychological and social pain, persons also experience severe spiritual pain [63-64]. Similarly, the feeling of hopelessness and stress experienced during mobbing can be equated to a confrontation with death [30], all the more so that the insecurity associated with job loss often becomes a reason for suicide [65]. In other words, in addition to explanations of the influence of some cultural and social variables on religious-spiritual help-seeking [19], the severity of mobbing experience can be understood as a stimulus to seek assistance that would meet the specific needs of the religious victim. This broadens the understanding of the motives for seeking religious-spiritual assistance and at the same time shows that in addition to assurance of psychological assistance, victims of mobbing may also benefit from support consistent with the individual’s religious beliefs.”
R2: Qualitative case studies would illustrate the various stresses placed on individuals, and strategies engaged in by the religious counsellors. Without this information, I don’t find this article helpful in understanding the nature of group bullying within the culture of Lithuania, the nature of interventions possible, and their effectiveness.
Authors’ answer (4). As it has already been mentioned, it was not aimed to investigate the work of religious advisors, the nature of interventions, which would surely benefit from a qualitative study. In this case, only a quantitative research approach was applied, as we sought to statistically test certain assumptions. Also, considering your comment, we are supplementing the part of limitations of the research:
“This study is quantitative and shows general trends, but some motives of employees seeking assistance may have remained unevaluated. These could be detailed by in-depth interviews with persons experiencing negative behaviour of co-workers. The results of the study show that non-religious victims of mobbing may also consider the opportunity to seek assistance from religious organizations. The qualitative research would help to understand how such intentions may be related to, for example, contextual actions such as living among religious persons, availability and quality of psychological assistance, general trust in therapy, societal attitudes related to psychological assistance, etc. Interventions applied in pastoral practice should also be evaluated.”
R2: In what kind of agency (e.g. public or private) does group bullying occur, and to what extent, and why? If the authors have any information in this regard, they should present it. How many respondents belonged to religious minorities (e.g. as Jews, Muslims, Jehovah’s Witnesses etc) – were they particularly likely to experience group bullying?
Authors’ answer (5). Because in this study we sought to find out why victims with different negative experiences tended to seek religious assistance in general, we tried not to overload with data that was not directly related to the questions raised. Representatives of religious minorities did not form homogenous groups; therefore, it was not possible to statistically process such data.
R2: A questionnaire is mentioned: the English-language translation should be appended. What are the “values” held by respondents? This could be a crucial concept, which is not elaborated. How many people who are “religious” but who rarely attend church, hold positive values regarding self and others? Can they be co-opted as resource people to counter workplace bullying? How do employers/directors regard intervention in the workplace by religious agencies?
Authors’ answer (6). The full version of the questionnaire was published in a previous article, which we refer to in the manuscript: Vveinhardt, J.; Deikus, M. Search for spiritual assistance in religious organizations: what are the motives of persons who have experienced destructive relationships at work?. Frontiers in Psychology 2021, 12, 702284.
The questionnaire is available at: https://www.frontiersin.org/articles/10.3389/fpsyg.2021.702284/full#supplementary-material
R2: My conclusion is that merely as a review of literature, and as a descriptive survey of religious beliefs and experience of workplace bullying, this article does not offer much advance on previous studies. We need to know more exactly how religious agencies and counsellors can intervene on behalf of bullied individuals, and what the outcomes have been, or are likely to be.
Authors’ answer (7). Thank you for your opinion.

Round 2
Reviewer 1 Report
Thank you for the opportunity to review this manuscript again. I appreciate the authors addressing my previously described concerns, and feel that this version of the manuscript is acceptable for publication.
Reviewer 2 Report
The authors' clarifications and additions have been helpful, and I now understand the purpose of the research more fully. While it remains a rather unexciting paper, it is sufficiently adequate to offer researchers and practitioners in this field an additional possibility for workplace interventions, and counselling referral.